# Comparing the Lower-Limb Muscle Activation Patterns of Simulated Walking Using an End-Effector-Type Robot with Real Level and Stair Walking in Children with Spastic Bilateral Cerebral Palsy

**DOI:** 10.3390/s23146579

**Published:** 2023-07-21

**Authors:** Yongjin Ahn, Juntaek Hong, Dain Shim, Joong-on Choi, Dong-wook Rha

**Affiliations:** Department and Research Institute of Rehabilitation Medicine, Severance Rehabilitation Hospital, Yonsei University College of Medicine, 50-1 Yonsei-ro, Seodaemun-gu, Seoul 03722, Republic of Korea; ahnmedic1@yuhs.ac (Y.A.); ghdwnsxor@yuhs.ac (J.H.); sdi3807@yuhs.ac (D.S.); joongonreh@yonsei.ac.kr (J.-o.C.)

**Keywords:** cerebral palsy, gait analysis, robotics, surface electromyography, rehabilitation

## Abstract

Cerebral palsy is a neurologic disorder caused by lesions on an immature brain, often resulting in spasticity and gait abnormality. This study aimed to compare the muscle activation patterns of real level and stair walking with those of simulated walking using an end-effector-type robot in children with spastic cerebral palsy. The electromyographic activities of the vastus lateralis, biceps femoris, tibialis anterior and medial gastrocnemius of nine children with spastic bilateral cerebral palsy were measured during gait using a wireless surface EMG device. Morning walk was used for the simulated gait. Differences in the muscle activation patterns between the real and simulated gait conditions were analyzed. In the loading response, all four muscles showed reduced activity during two simulated conditions. In mid-stance, mGCM showed reduced activity during simulated conditions, whereas BFem showed greater activity during simulated level walking. In the swing phase, BFem and TAnt activity was reduced during the simulated conditions. The onset–offset of the VLat, BFem and TAnt activity was significantly delayed during simulated versus real level walking. No differences in activity onset–offset were observed between the simulated level and stair conditions. In conclusion, the robot-simulated gait showed differences in its muscle activation patterns compared with the real gait conditions, which must be considered for gait training using an end-effector-type robot.

## 1. Introduction

Cerebral palsy is a non-progressive neurologic disorder caused by damage to an immature brain during the antenatal, perinatal or postnatal period [1,2]. It can manifest in a variety of movement disorders, which contribute to abnormalities in gait [3,4,5]. Gait disturbance is a prominent complication of cerebral palsy that adversely affects patients’ functional outcome and quality of life [6,7,8], and there have been attempts to train patients to overcome the difficulties faced during level and stair walking [9,10,11]. As gait kinematics and the required torques of various joints differ between level walking and climbing stairs [12,13], gait training on a level surface cannot adequately prepare patients for stair climbing; therefore, training patients on stairs in addition to level surface training is essential. However, stair training presents numerous obstacles for patients, with the risk of falling and subsequent injury being major concerns [14]. End-effector-type robot-simulated stair climbing is an alternative to labor-intensive conventional training, in which therapists manually assist the patients. However, whether muscle activation patterns during simulation using an end-effector-type robot are different from real-life gait or not has not been studied enough; this is crucial in determining the efficacy of robotic-simulated training. The muscle activation patterns of hemiplegic stroke patients during gait simulations using an end-effector-type robot were reported to be closer to those of healthy subjects compared to the real-life gait training, showing that the shank muscles were activated in a more ‘timely, correct’ fashion during the simulated gait [14]. However, the effect of robot-simulated gait on muscle activation patterns has not yet been studied in children with bilateral spastic cerebral palsy. Thus, this study aimed to analyze the differences in the muscle activation patterns among children with bilateral spastic cerebral palsy during real level walking and stair-climbing conditions compared with simulated walking using an end-effector-type gait-training robot. In case differences in the muscle activation patterns between the real-life and simulated training conditions are observed, further studies can be performed to confirm the clinical significance of such differences.

## 2. Materials and Methods

### 2.1. Participants

Nine children (six boys and three girls, mean [standard deviation] age: 9 years 4 months [2 years 6 months]) with spastic bilateral cerebral palsy undergoing regular outpatient follow-up in the department of pediatric rehabilitation at a university-affiliated tertiary-care hospital participated in this study (Table 1). Two medical doctors specializing in pediatric rehabilitation, each with more than 20 years of clinical experience, made the confirmative diagnosis of spastic bilateral cerebral palsy according to the diagnostic workup recommended by the American Academy of Neurology [2], and all of the recruited participants had brain imaging showing evidence of brain lesions. The sampling frame included all children that visited the outpatient clinic of the pediatric rehabilitation department from January 2022 to March 2022. The inclusion criteria were as follows: (a) children with spastic bilateral cerebral palsy between the ages of 5 and 13; (b) Gross Motor Function Classification System (GMFCS) level I or II; (c) height > 100 cm; and (d) ability to follow the study protocols. GMFCS is a classification system that categorizes patients with cerebral palsy into 5 levels based on their gross motor function. Only GMFCS level I or II were included, as levels below II cannot walk independently on a level surface.

The exclusion criteria were as follows: (a) inability to obey the study protocols; (b) other types of cerebral palsy (e.g., dystonic or ataxic); (c) severe lower-limb spasticity (Modified Ashworth Scale ≥ 3) or severe fixed-joint contractures (above 20°) that hinder the operation of the robot and/or fastening the foot to the robot’s foot-plate; (d) history of botulinum toxin injection or lower-limb cast application within 3 months prior to the study; (e) history of lower-limb orthopedic surgery within 6 months prior to the study; (f) skin defects precluding the use of surface electromyography (EMG); and (g) the presence of other musculoskeletal diseases (e.g., myopathy, osteomalacia). The Modified Ashworth Scale (MAS), a 5-point scale used for the measurement of spasticity, is the scale most widely used clinically [15]; a score of 3 or above is defined as difficulty in the passive movement of a joint, indicating severe spasticity [16]. Two children had histories of orthopedic surgeries (Table 1). We included patients who had undergone orthopedic surgeries as there has been a report stating that muscle activation patterns during gait are not significantly altered following calf muscle surgery [17]. Before electromyographic assessment, the participants’ MAS scores for their bilateral ankle plantarflexors, hamstring, and hip adductor muscles, as well as their height, weight, and Gross Motor Function Measure 88 score (GMFM 88) were measured (Table 1). GMFM 88 is a scale with a maximum of 100 points used to assess five domains of the functional capabilities (i.e., lying and rolling; sitting; crawling and kneeling; standing; walking, running and jumping) of children with cerebral palsy [18]. This study was registered for clinical trial (registration number: PRE20220902-001).

### 2.2. Devices

Morning Walk (CUREXO, Co., Ltd., Seoul, Republic of Korea) is a lower-limb end-effector-type rehabilitation robot developed in 2014, and is widely used in clinical practice for the rehabilitation of patients with a variety of neuromuscular or skeletal disorders, ranging from stroke [19] to post-knee surgery [20]. In contrast to exoskeleton-type robots, end-effector-type robots are connected to patients at distal interfaces, allowing free motion in proximal joints as only the feet are bound to the metal plates [21]. The foot plates of the robot move in pre-determined trajectories that simulate either level or stair walking (Figure 1). The robot’s trajectories were derived from the foot trajectories of healthy adults’ level and stair-climbing gaits using a 3D motion-capture system, similar to the methodology adopted for other end-effector-type robots [14]. Gait parameters such as the gait speed, step length and height, initial contact angle at the stance phase, and toe-off angle at the start of the swing phase are adjustable in accordance with patients’ needs. For safety, the robot is equipped with a safety belt and chest support with a Velcro strap.

### 2.3. Assessment Protocol

The surface EMG data were acquired during the following four conditions: walking on a level surface at a self-selected speed (RLevel); climbing a flight of stairs with 10 steps with a riser height of 14.0 ± 0.5 cm and a tread depth of 30.0 ± 1.0 cm, also at a self-selected speed and in an alternating fashion (using the handrail for balance control was allowed if necessary) (RStair); level walking simulation on the robot at a comfortable cadence (SLevel); and stair-climbing simulation on the robot also at a comfortable cadence (SStair) (Figure 2). These four gait conditions were executed in a random order in a single session. The duration of assessment for each condition was 30 s, and the children were given a 5 min rest between trials to minimize fatigue and provide adequate wash-out periods.

Before surface EMG data were acquired during the robot-simulated gait conditions, the children were given 10 min to familiarize themselves with the robot and choose a comfortable cadence. During the robotic simulation of level walking and stair climbing, the foot plate-to-floor angle of initial contact was set to 5 degrees and that of the toe-off was set to −30 degrees to enable comfortable walking [22,23]. The step length for the simulated level and stair walking was fixed to 30 cm to match the tread depth of the real staircase. The simulated step height was fixed to 14 cm to match the riser height of the real staircase.

### 2.4. Electromyography Measurement

Surface EMG was measured using the Delsys Trigno system (Delsys Inc., Boston, MA, USA), which consists of portable wireless EMG sensors with four bar electrodes that are attached to the skin directly above the muscle to be examined. Surface EMG sensors record the summated action potentials produced by the muscle fibers of a contracting muscle. Each sensor recorded 3-axis gyroscope and accelerometer data and time data in synchrony with the EMG data.

Four EMG channels were used, each designated to one of the four lower-limb muscles of the more affected limb (VL, BF, TA and medial GAST) according to SENIAM guidelines. An additional sensor was placed at the medial malleolus of the examined limb to record the accelerometer data for gait phase detection. The EMG channels were synchronized in time, facilitating analysis with reference to the gait phases.

### 2.5. Data Analysis

Signals from the sensors were streamed to a laptop via Bluetooth connection, and were recorded using Delsys EMGworks software with recording frequencies of 1111.11 Hz for EMG channels and 148.15 Hz for accelerometer channels. The gait phase was detected using vertical accelerometer signals (ACC-Z) from the sensor attached to the medial malleolus of the examined limb. The time of initial contact was determined by detecting peaks in the accelerometer signals. One representative gait cycle was chosen for each gait condition for analysis and time-normalized, such that the cycle duration was set to 100%. Each gait cycle was divided into 10 equal phases for further analysis [24]. The transition from the stance phase to swing phase occurred from 62 to 75% of the gait cycle for real-life level walking, and from 62 to 70% of the gait cycle for real-life stair climbing. For the robotic simulated gait, the transition from stance to swing was also set at the 6th phase of the gait cycle.

In the first step, the EMG signals were band-pass filtered between 20 and 450 Hz using 2nd-order Butterworth filters. The EMG signals for each gait cycle were processed using a 10-point root mean square (RMS) algorithm, such that each RMS value represented 1/10 of a gait cycle. The RMS values were analyzed as they are commonly used to assess the muscle activation level [25,26,27,28]. The RMS values were normalized to percentages of maximal contraction during real level walking (%MCRL) such that the maximum RMS value of a muscle from the RLevel condition was set to 100% MCRL.

In the second step, the onset–offset points of muscle activation were determined using the threshold-based method [29,30] in conjunction with the Teager–Kaiser energy operator (TKEO) signal conditioning algorithm [31,32] (Figure 3). The conditioned signal was then used for onset–offset determination using the threshold-based method.

### 2.6. Statistical Analysis

Statistical analysis was performed using R version 4.2.2 software (R Foundation for Statistical, Computing, Vienna, Austria). In order to compare muscle activities between the two gait conditions, 10% MCRLs of a muscle at each gait phase were compared using the Wilcoxon signed-rank test. The onset–offset of muscle activities between the two gait conditions were compared using the Wilcoxon signed-rank test.

The sample size was calculated using G* Power (Universität Düsseldorf, Germany), considering the mean activity of each muscle as the primary outcome measure and using the Wilcoxon signed-rank test between the two gait conditions with a type I error of 0.05, power of 0.80, and effect size of 1.00. The effect size was calculated using data from a previous study comparing the muscle activity of hemiparetic patients during treadmill gait and an end-effector-type robot-simulated gait [33]. The sample sizes of previous studies with a similar design [14,33,34,35] were also consulted, which ranged from 6 to 14.

## 3. Results

### 3.1. Degree of Muscle Activation in Each Gait Phase

The degree of muscle activation, as represented by %MCRL, was lower for all four muscles during the loading response in the SLevel condition than in the RLevel condition (Table 2, Figure 4). The BF and TA showed significantly lower activity in phase 1 (both *p* = 0.004) of the gait cycle. In the mid to terminal stance, the BF showed higher activity in phases 4–6 (*p* = 0.012, 0.039, 0.008, respectively), whereas the medial GAST showed lower activity in phase 3 (*p* = 0.020) during the SLevel condition. The BF showed lower activity in phase 10 during the terminal swing in the SLevel condition (*p* = 0.004).

Comparing the muscle activity between the Rstair and Sstair conditions revealed similar patterns, as all of the examined muscles showed significant reductions in activity during loading response in the simulated condition compared with the real condition: the VL and medial GAST showed lower activity in phase 2 (*p* = 0.020, 0.040, respectively), and the BF and TA showed lower activity in phase 1 (*p* = 0.020, 0.004, respectively). During terminal stance to swing, the medial GAST showed lower activity during simulation in phases 5–7 (*p* = 0.012, 0.008, 0.020, respectively). During terminal swing, the BF showed lower activity in phase 10 (*p* = 0.020) and the TA showed lower activity in phases 9 and 10 (*p* = 0.012, 0.012, respectively).

### 3.2. Onset–Offset of Muscle Activity

Comparing the onset–offset of muscle activity between the Rlevel and Slevel conditions revealed that the VL had a significantly delayed offset during simulation (*p* = 0.036), whereas the BF and TA exhibited the prolongation of both onset (*p* = 0.009, 0.030, respectively) and offset (*p* = 0.009, 0.014, respectively) during simulation (Table 3, Figure 5).

Interestingly, no statistically significant differences were found in the onset–offset of activity in all of the examined muscles between the RStair and SStair conditions (Table 3, Figure 5), nor between the SLevel and SStair conditions (Table 3, Figure 6).

## 4. Discussion

Surface EMG data from the four lower-limb muscles of the nine participants were acquired during real and simulated gait conditions. All of the recruited participants successfully completed the assessment protocol without adverse events, including during the use of Morning Walk.

### 4.1. Comparison of Muscle Activation Patterns between Simulated and Real-Life Gait Conditions

During the loading response physiological level gait, the ankle dorsiflexors, knee extensors and hip extensors are activated. These muscle groups are also activated to accept weight during stair climbing [36]. In our study, the %MCRLs of the VL, BF and TA muscles were decreased during loading response in the simulated gait conditions (SLevel, SStair) compared with the real-life gait conditions (RLevel, RStair). During simulated walking, the children’s feet were in constant contact with the foot plates; therefore, the ground reaction force may not have been generated by the impact of initial contact following the swing phase. This lack of a definite transition from swing to stance may have reduced the activation of muscles that stabilize the lower limb during initial contact and loading response. A previous study comparing real and end-effector-type robot-simulated gait in healthy persons reported similar patterns [35].

The mid to terminal stance phases of physiological level and stair walking are marked by the increasing activity of the ankle plantar flexors [36,37]. During simulated conditions (SLevel, SStair), the medial GAST generally showed reduced activity compared with real conditions (RLevel, RStair), which aligns with previous studies [34,35]. In the simulated gait, as one limb underwent the mid to terminal stance, the contralateral limb was still fixed to the foot plate of the robot, deviating from real-life gait conditions. Therefore, the mid to terminal stance phase in the simulated gait did not exactly constitute single-limb support, as the opposite limb partially bore weight. This may have reduced the external moment dorsiflexing of the ankle, leading to decreased medial GAST activity. Also, similar patterns of reduced medial GAST activity were seen at the end of the stance phase (i.e., push-off) during the SStair condition compared with the RStair condition. Previous studies of healthy persons similarly revealed the lower activity of the gastrocnemius during simulated gait when using either exoskeleton or end-effector-type robots [34,35], both of which provided direct assistance to the children’s ankle motion. The demands on the gastrocnemius were thought to be alleviated because the foot plates assisted ankle plantarflexion during push-off. In contrast to the ankle plantar flexors, the BF showed greater activity in the mid to terminal stance phases (phases 4–6) during the SLevel condition compared with the RLevel condition. This aligns with a previous study that showed greater BF activity during the stance phase in simulated level gait conditions in non-ambulatory hemiparetic stroke patients [33]. In this study, the affected limb followed a more physiologic trajectory during the robot-simulated gait, with a greater range of motion (ROM) in the hip sagittal due to the robot’s highly symmetrical movement. Such an increase in the ROM of the affected hip during the simulated gait could elongate the external moment arm, resulting in increased BF activity. Although we did not measure hip joint kinematics, a similar increase in the ROM of the hip may have occurred during the simulated gait in our study, which could explain the increased BF activity.

During the swing phase, the ankle dorsiflexors [37] are activated to ensure foot clearance during physiological level gait, and during the terminal swing, the increased activity of the hamstring muscles decelerates the forward motion of the lower-leg segment during knee extension [38]. Similar activation patterns in the ankle dorsiflexors and hamstring muscles can be seen during physiological stair climbing [36,38]. The TA generally showed lower activity in the swing phase during the simulated conditions (SLevel, SStair) compared with the real-life gait conditions (RLevel, RStair), aligning with previous studies [34,35]; however, this reduction in activity was statistically significant only when comparing the RStair and SStair conditions. As the foot plates supported the ankle throughout the entire swing phase, the demand on the TA to dorsiflex the ankle was alleviated during this phase. The difference in the activity of the TA between real and simulated conditions may be accentuated in stair conditions (RStair, SStair) due to greater demand being placed on the ankle dorsiflexors during the swing phase of real stair climbing (RStair) compared with level walking (RLevel) in order to achieve adequate foot clearance. Although the ankle plantar flexors are known to be silent during the swing phase in physiological level and stair walking [36,37], the medial GAST showed peak activity at the transition from the stance to swing phase during the RStair condition and remained active during significant portions of the swing phase in our study. This might be due to the characteristic prolongation of activity and the co-activation of antagonist muscles during gait in patients with cerebral palsy [39,40,41,42]. Direct ankle assistance provided by simulated stair walking (SStair) mitigated this effect, resulting in a more physiological pattern, as was also noted by Hesse et al. [14]. The BF showed lower activity in the terminal swing phase of the simulated gait (SLevel, Sstair) compared with the real-life gait (RLevel, RStair). During the simulated gait, the swing phase limb passively followed the pre-determined trajectory of the feet bound to the robot’s foot plates, which may have reduced the burden placed on the BF to decelerate the lower leg, resulting in decreased activation.

### 4.2. Comparison of Onset–Offset of Muscle Activity

The onset–offset of activity was delayed and prolonged in the VL, BF and TA during the SLevel condition compared with the RLevel condition. Previous studies have similarly reported delays and the prolongation of muscle activity during robot-simulated gait [14,35], which has been attributed to a softer impact during initial contact, and tibial advancement being unimpeded by metatarsal joints [14]. Our study also found no significant difference in the onset and offset points between the RStair and SStair conditions (Table 3, Figure 5), further supporting the findings of Hesse et al. [14].

Although we observed the prolongation of onset–offset during the RStair condition compared with the RLevel condition, this difference was not present between the SStair and SLevel conditions (Table 3, Figure 6). We also found no major differences between the 10% MCRLs within a gait cycle between the two simulated conditions (SLevel vs. SStair) (Table 4). Although real and simulated stair climbing (RStair, SStair) required greater hip and knee muscle power compared with level walking (RLevel, SLevel) [43], the foot plates supported the vertical motion of the children’s limbs during the simulations. As such, additional muscle forces might not be needed to achieve greater hip and knee ROM during simulated stair climbing (SStair) compared with simulated level walking (SLevel) even though the robot’s foot plates follow the trajectory of stair climbing. This result shows the limitations of the robot’s ambulation modes in eliciting distinct differences in lower-limb muscle activity.

### 4.3. Limitations

We did not consider the heterogeneity of the children’s gait patterns, GMFCS levels, age and orthopedic surgical history when recruiting for this study. Further research analyzing children with homogenous gait patterns, clinical and medical characteristics may help clarify the results.

Second, we did not compare the joint kinematics between real-life and simulated gait conditions. Therefore, it was difficult to judge whether the differences in the muscle activation patterns were caused by the different joint kinematics of the robot-simulated gait, or whether the kinematics were well simulated, and the differences in muscle activation were due to the assistance provided by the robot.

Third, as our primary goal was to confirm whether the electromyographic pattern at the time of training was different between real and simulated conditions, we did not inspect the therapeutic effects of robot-assisted training. Further clinical study is needed to confirm whether the muscle activation patterns change pre- and post- intervention, and to demonstrate the clinical significance of the difference in the muscle activation patterns between the real-life and simulated gait conditions shown in this study.

In addition, the EMG of the hip flexor muscles was not analyzed, as the iliopsoas was too profound to be measured using the surface electrodes and the activity of the rectus femoris was difficult to distinguish from that of the adjacent vasti muscles [44]. Thus, this study’s view of swing-phase muscle activity was limited.

## 5. Conclusions

Muscle activities were generally reduced in the simulated gait versus real-life gait conditions. This result is ascribable to some inherent characteristics of end-effector-type robot-assisted gait simulation, i.e., the absence of impact during the transition from swing to stance, the passive assistance of ankle plantar flexion during push-off and dorsiflexion during the swing phase, and the passive acceleration and deceleration of thigh and shank segments.

The examined muscles showed the prolonged onset–offset of activity during simulated walking compared with real level walking. The two simulated gait modes (level and stair climbing) of the end-effector-type robot did not elicit significant differences in muscle activation patterns and timing. Since only an end-effector-type robot was tested in this study, the results cannot apply to robots with different mechanisms, such as exoskeleton-type robots. Further studies are needed to correlate the electromyographic patterns with clinical effects and to make relevant clinical recommendations.

## Figures and Tables

**Figure 1 sensors-23-06579-f001:**
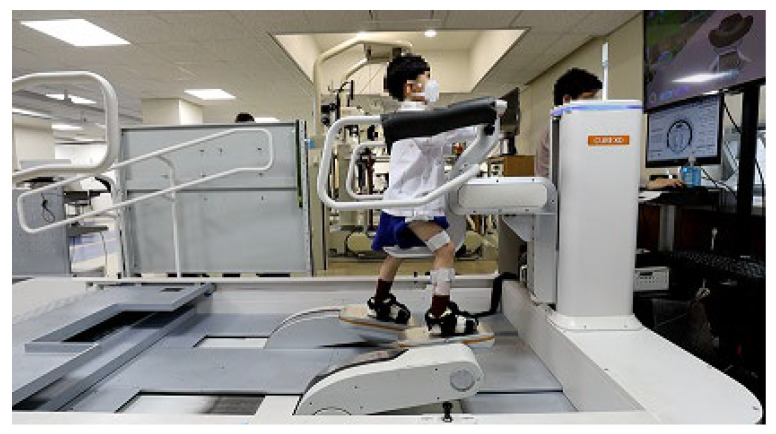
Photograph of a child with cerebral palsy participating in end-effector-type robotic-simulated gait training using Morning Walk (CUREXO, Co., Ltd., Seoul, Republic of Korea) with portable wireless electromyography sensors on the vastus lateralis (VL), biceps femoris (BF), tibialis anterior (TA) and medial gastrocnemius (medial GAST) of the more affected limb.

**Figure 2 sensors-23-06579-f002:**
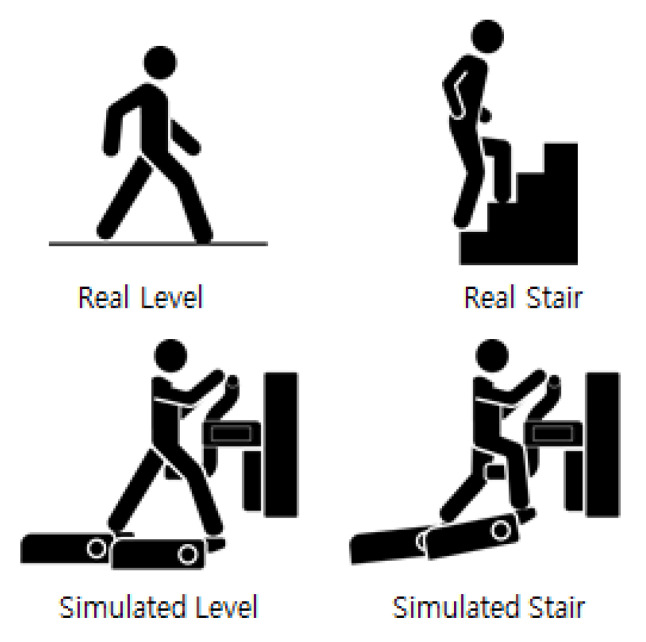
The four gait conditions.

**Figure 3 sensors-23-06579-f003:**
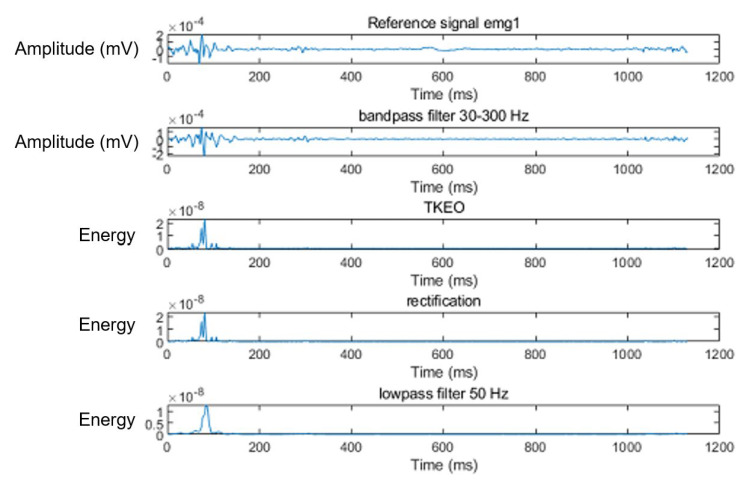
Determination of onset–offset points of electromyographic signal. Reference electromyographic signal was conditioned using band-pass filtering at 30–300 Hz (6th-order Butterworth filter), a Teager–Kaiser energy operator, and was rectified and low-pass filtered at 50 Hz (2nd-order Butterworth filter). Mean and standard deviation of the baseline signal after the conditioning was calculated, from which the threshold was determined as T = (mean) + 15 × (standard deviation). The onset and offset time were estimated as the point at which the conditioned curves crossed the set threshold.

**Figure 4 sensors-23-06579-f004:**
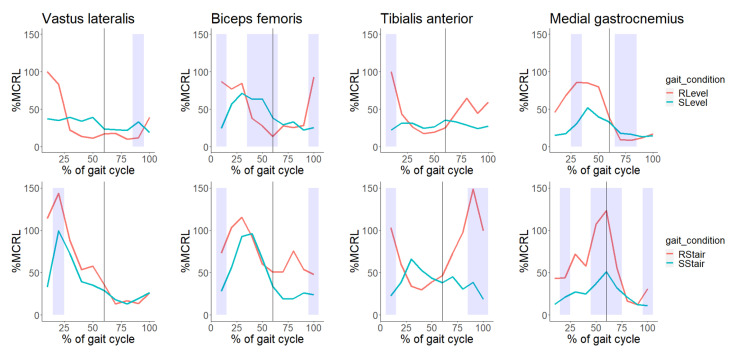
Median %MCRL (% of maximal contraction during real level walking) of each of the examined muscles (vastus lateralis, biceps femoris, tibialis anterior and medial gastrocnemius) during real level (Rlevel) walking and simulated level (Slevel) walking on the robot, and during real stair (Rstair) walking and simulated stair (Sstair) walking on the robot. Phases of the gait cycle showing statistical significance between the %MCRLs of the two gait conditions are shaded in light blue. Vertical lines are placed at 60% of the gait cycle on each chart to represent the transition from the stance to swing phase.

**Figure 5 sensors-23-06579-f005:**
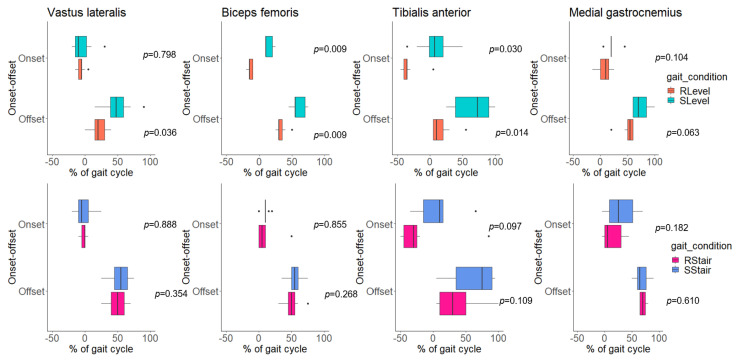
Boxplots of onset–offset points (% of gait cycle) of electromyographic activity of each of the examined muscles (vastus lateralis, biceps femoris, tibialis anterior and medial gastrocnemius) during real level (RLevel) walking and simulated level (SLevel) walking on the robot and during real stair (RStair) walking and simulated stair (SStair) walking on the robot. *p*-values of the Wilcoxon signed-rank test between the two gait conditions are expressed on each plot.

**Figure 6 sensors-23-06579-f006:**
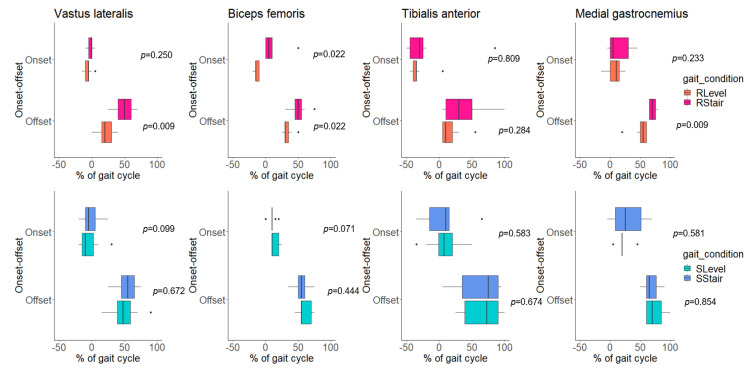
Boxplots of onset–offset points (% of gait cycle) of electromyographic activity of each of the examined muscles (vastus lateralis, biceps femoris, tibialis anterior and medial gastrocnemius) during real level (RLevel) and real stair (RStair) walking, and during simulated level (SLevel) and simulated stair (SStair) walking on the robot. *p*-values of the Wilcoxon signed-rank test between the two gait conditions are expressed on each plot.

**Table 1 sensors-23-06579-t001:** Clinical data of participants.

Participants	Age	Gender	Etiology	GMFCS Level	Ankle PF Spasticity *		Hamstring Spasticity *		Hip Adductor Spasticity *		Height (cm)	Weight (kg)	GMFM 88 Score	More Involved Side
					Right	Left	Right	Left	Right	Left				
1 †	9Y6M	Male	PVL	II	1	2	1	1	1	1	137.0	37.4	72.0	Left
2	5Y9M	Male	PVL	I	1	1	0	0	0	0	110.6	18.7	98.9	Right
3	8Y10M	Male	Pachygyria, polymicrogyria	I	1	1	0	0	0	0	124.7	23.4	99.7	Right
4	7Y7M	Female	PVL	II	1	1+	0	1	0	0	126.7	34.8	96.0	Left
5 †	11Y11M	Male	PVL	I	1	1	0	0	0	0	156.8	58.9	87.9	Right
6	6Y	Female	PVL	II	1	1+	1	1	1	1	108.8	15.1	89.7	Left
7	12Y1M	Male	PVL	I	1	1	0	0	0	0	146.1	38.0	99.2	Left
8	10Y1M	Male	PVL	I	1	1	0	0	0	0	142.1	31.5	91.2	Left
9	12Y4M	Female	PVL	I	1	1	0	0	0	0	156.0	62.0	-	Right

PVL: periventricular leukomalacia, PF: Plantarflexor, GMFM 88: Gross Motor Function Measure 88 scale, * Spasticity was measured according to the Modified Ashworth scale (MAS), † Participants who underwent orthopedic surgery. Participant 1 underwent a distal-femur-extension-shortening osteotomy and medial hamstring lengthening on the right leg, a distal-femur-extension-shortening derotational osteotomy on the left leg and gastrocnemius soleus recession on both legs 8 months prior to enrolment. Participant 5 underwent a calcaneal-lengthening osteotomy, peroneus brevis lengthening and heel cord lengthening on the right leg, and gastrocnemius soleus recession on the left leg 16 months prior to enrolment.

**Table 2 sensors-23-06579-t002:** Comparison of activities of the four lower-limb muscles during simulated (SLevel, SStair) versus real (RLevel, RStair) gait conditions.

	Comparison of %MCRL (%, median [interquartile range]) of the VL between simulated versus real gait conditions									
	phase 1	phase 2	phase 3	phase 4	phase 5	phase 6	phase 7	phase 8	phase 9	phase 10
RLevel	100.0 [79.7, 100.0]	83.0 [58.0, 100.0]	22.1 [17.0, 39.9]	13.6 [10.0, 20.7]	11.2 [8.8, 19.1]	16.9 [11.3, 17.6]	17.9 [7.3, 19.7]	10.0 [8.6, 16.2]	11.6 [9.9, 12.6]	39.2 [29.5, 72.7]
Slevel	37.4 [11.7, 113.2]	35.0 [14.6, 74.4]	39.0 [20.4, 54.2]	33.9 [20.0, 45.5]	39.0 [17.3, 41.1]	23.8 [21.9, 40.1]	22.7 [11.7, 29.0]	21.7 [11.4, 26.8]	33.0 [13.1, 111.7]	18.7 [11.7, 88.8]
*p*-value	0.359	0.250	0.301	0.055	0.203	0.570	0.734	0.074	0.027 †	0.652
Rstair	113.8 [52.5, 171.5]	143.5 [75.5, 212.9]	88.0 [28.2, 103.5]	53.4 [29.9, 87.0]	57.7 [20.8, 84.0]	35.9 [22.8, 61.8]	13.1 [10.6, 28.6]	16.6 [14.4, 22.0]	13.3 [11.7, 27.7]	26.0 [12.7, 57.4]
Sstair	32.9 [20.2, 148.0]	99.4 [34.3, 146.2]	72.1 [30.5, 119.6]	39.4 [26.0, 105.4]	35.2 [11.8, 84.7]	28.6 [11.8, 54.9]	17.9 [11.4, 28.9]	12.8 [10.2, 21.4]	19.3 [11.3, 29.5]	26.3 [12.0, 58.1]
*p*-value	0.074	0.020 †	0.652	0.910	0.570	0.570	0.820	0.570	0.301	0.164
	Comparison of %MCRL (%, median [interquartile range]) of the BF between simulated versus real gait conditions									
	phase 1	phase 2	phase 3	phase 4	phase 5	phase 6	phase 7	phase 8	phase 9	phase 10
Rlevel	86.7 [61.3, 100.0]	77.0 [49.6, 99.0]	84.5 [55.7, 91.9]	37.6 [18.5, 41.1]	27.8 [16.0, 36.6]	13.5 [11.7, 29.8]	27.7 [19.0, 38.9]	25.6 [8.4, 28.8]	28.6 [21.0, 52.8]	92.8 [63.3, 96.3]
Slevel	24.6 [15.9, 39.6]	56.6 [44.4, 65.2]	71.0 [53.7, 85.4]	63.2 [60.7, 92.7]	63.9 [48.2, 73.6]	38.1 [25.4, 47.9]	29.4 [15.4, 36.9]	33.2 [19.4, 35.4]	22.1 [17.1, 27.7]	25.5 [20.6, 32.5]
*p*-value	0.004 †	0.164	0.734	0.012 †	0.039 †	0.008 †	1.000	0.250	0.055	0.004 †
Rstair	73.2 [70.0, 128.0]	103.7 [86.9, 134.2]	115.1 [63.8, 142.7]	92.3 [75.6, 147.5]	60.2 [58.2, 101.5]	50.6 [25.8, 113.5]	51.1 [43.9, 108.4]	75.5 [32.7, 89.5]	53.9 [34.6, 65.4]	47.4 [40.4, 63.6]
Sstair	27.8 [24.7, 37.9]	56.2 [48.4, 98.1]	92.6 [64.4, 121.3]	96.2 [68.1, 113.2]	66.2 [55.3, 83.7]	33.4 [24.5, 55.6]	18.9 [15.6, 40.1]	19.1 [13.0, 38.5]	25.9 [19.6, 44.1]	23.7 [20.1, 27.8]
*p*-value	0.020 †	0.098	0.910	1.000	0.910	0.164	0.055	0.250	0.429	0.020 †
	Comparison of %MCRL (%, median [interquartile range]) of the TA between simulated versus real gait conditions									
	phase 1	phase 2	phase 3	phase 4	phase 5	phase 6	phase 7	phase 8	phase 9	phase 10
Rlevel	100.0 [89.7, 100.0]	43.2 [35.1, 52.7]	25.9 [21.2, 34.0]	17.5 [12.3, 30.2]	19.2 [16.6, 31.0]	24.9 [15.5, 34.8]	45.5 [37.7, 80.9]	64.5 [36.3, 73.7]	44.2 [19.3, 51.2]	59.2 [20.9, 76.9]
Slevel	22.2 [18.2, 34.6]	31.9 [18.5, 64.2]	31.1 [22.7, 57.1]	24.6 [10.9, 47.7]	26.6 [14.9, 61.2]	35.2 [13.7, 53.8]	33.1 [23.0, 44.8]	28.9 [22.8, 58.3]	24.3 [17.4, 79.3]	27.5 [15.8, 60.1]
*p*-value	0.004 †	0.910	0.359	0.301	0.426	0.098	0.250	0.301	0.496	0.164
Rstair	103.3 [64.7, 149.4]	59.8 [33.3, 76.5]	33.8 [23.3, 71.5]	29.6 [18.1, 77.8]	38.9 [28.7, 45.1]	46.2 [43.9, 56.6]	72.8 [30.0, 75.4]	97.9 [81.8, 129.2]	148.6 [126.1, 155.5]	99.4 [92.1, 158.8]
Sstair	22.5 [19.5, 25.5]	38.6 [26.2, 64.5]	66.0 [39.0, 105.9]	52.8 [36.8, 79.5]	43.6 [11.5, 63.2]	38.2 [10.7, 60.2]	45.2 [30.0, 73.2]	30.6 [17.1, 77.9]	38.6 [11.7, 65.9]	18.2 [12.0, 29.50]
*p*-value	0.004 †	0.129	0.496	0.496	1.000	0.250	0.429	0.129	0.012 †	0.012 †
	Comparison of %MCRL (%, median [interquartile range]) of medial the GAST between simulated versus real gait conditions									
	phase 1	phase 2	phase 3	phase 4	phase 5	phase 6	phase 7	phase 8	phase 9	phase 10
Rlevel	46.0 [17.7, 83.7]	68.0 [29.8, 72.7]	85.8 [48.2, 92.3]	84. [74.3, 100.0]	79.5 [56.4, 91.1]	39.2 [15.0, 45.5]	9.3 [7.4, 10.2]	8.7 [4.7, 13.3]	12.0 [4.6, 17.8]	17.1 [5.9, 37.0]
Slevel	15.3 [11.4, 23.8]	17.5 [9.3, 25.5]	30.8 [10.8, 72.2]	51.7 [10.7, 72.3]	39.6 [16.0, 67.4]	33.1 [17.1, 62.1]	18.1 [12.7, 36.1]	16.4 [13.1, 37.3]	13.2 [7.0, 24.2]	14.4 [12.3, 19.3]
*p*-value	0.074	0.074	0.020 †	0.055	0.164	0.910	0.012 †	0.020 †	0.359	0.652
Rstair	43.1 [12.1, 53.0]	44.0 [18.3, 56.5]	71.7 [21.7, 78.8]	57.6 [26.2, 81.7]	107.3 [65.0, 185.8]	123.2 [77.7, 186.3]	54.9 [45.9, 83.0]	16.2 [8.6, 24.1]	12.2 [6.3, 18.6]	30.8 [13.0, 43.1]
Sstair	12.4 [8.2, 30.9]	21.0 [10.5, 31.2]	27.0 [11.2, 32.6]	24.5 [8.1, 65.9]	37.3 [13.5, 64.4]	50.9 [17.9, 57.4]	31.6 [26.1, 41.8]	20.9 [10.8, 34.8]	12.0 [8.1, 34.6]	10.7 [7.1, 30.0]
*p*-value	0.055	0.040 †	0.074	0.164	0.012 †	0.008 †	0.020 †	0.820	0.734	0.040 †

† indicates *p*-value < 0.05, %MCRL: (% of maximal contraction during real level walking) (10 phases per gait cycle), VL: vastus lateralis; BF: biceps femoris; TA: tibialis anterior; medial GAST: medial gastrocnemius, Rlevel: real level walking condition, Slevel: simulated level walking on the robot, Rstair: real stair walking condition, Sstair: simulated stair walking on the robot, Wilcoxon signed-rank test results for %MCRL of each of the four examined muscles during Rlevel and Slevel, and during Rstair and Sstair. Data are presented as median [interquartile range].

**Table 3 sensors-23-06579-t003:** Comparison of onset and offset of activities of the four lower-limb muscles in the four different gait conditions.

Onset and Offset (% of Gait Cycle, Median [Interquartile Range]) of EMG Activity of Each Muscle								
	VL		BF		TA		Medial GAST	
	Onset	Offset	Onset	Offset	Onset	Offset	Onset	Offset
RLevel	−5 [−10, −5]	20 [15, 30]	−15 [−15, −10]	30 [30, 35]	−35 [−40, −35]	10 [5, 20]	10 [0, 15]	55 [50, 60]
SLevel	−10 [−15, 3]	48 [39, 59]	10 [10, 20]	55 [55, 70]	8 [−1, 20]	73 [39, 90]	20 [20, 20]	70 [60, 85]
*p*-value	0.798	0.036 †	0.009 †	0.009 †	0.030 †	0.014 †	0.104	0.063
RStair	0 [−5, 0]	50 [40, 60]	5 [0, 10]	50 [45, 55]	−30 [−45, −25]	30 [10, 50]	5 [0, 30]	70 [65, 75]
SStair	−5 [−10, 5]	55 [45, 65]	10 [10, 10]	55 [50, 60]	10 [−15, 15]	75 [35, 90]	25 [9, 51]	65 [60, 76]
*p*-value	0.888	0.354	0.855	0.268	0.097	0.109	0.182	0.610
RLevel	−5 [−10, −5]	20 [15, 30]	−15 [−15, −10]	30 [30, 35]	−35 [−40, −35]	10 [5, 20]	10 [0, 15]	55 [50, 60]
RStair	0 [−5, 0]	50 [40, 60]	5 [0, 10]	50 [45, 55]	−30 [−45, −25]	30 [10, 50]	5 [0, 30]	70 [65, 75]
*p*-value	0.250	0.009 †	0.022 †	0.022 †	0.809	0.284	0.233	0.009 †
SLevel	−10 [−15, 3]	48 [39, 59]	10 [10, 20]	55 [55, 70]	8 [−1, 20]	73 [39, 90]	20 [20, 20]	70 [60, 85]
SStair	−5 [−10, 5]	55 [45, 65]	10 [10, 10]	55 [50, 60]	10 [−15, 15]	75 [35, 90]	25 [9, 51]	65 [60, 76]
*p*-value	0.099	0.672	0.071	0.444	0.583	0.674	0.581	0.854

Tonic or silent muscle activity patterns were excluded from analysis, † indicates *p*-value < 0.05, VL: vastus lateralis; BF: biceps femoris; TA: tibialis anterior; medial GAST: medial gastrocnemius, RLevel: real level walking condition, SLevel: simulated level walking on the robot, RStair: real stair walking condition, SStair: simulated stair walking on the robot, Wilcoxon signed-rank test on onset–offset (% of gait cycle) of electromyographic (EMG) activity of each of the four examined muscles during RLevel and SLevel, during RStair and SStair, during RLevel and RStair, and during SLevel and SStair. The onset and offset of EMG activity were rounded to the nearest 5% of the gait cycle.

**Table 4 sensors-23-06579-t004:** Comparison of activities of the four lower-limb muscles during simulated level (SLevel) and stair (SStair) conditions.

	Comparison of %MCRL (%, median [interquartile range]) of the VL between simulated level versus stair gait conditions									
	phase 1	phase 2	phase 3	phase 4	phase 5	phase 6	phase 7	phase 8	phase 9	phase 10
SLevel	37.4 [11.7, 113.2]	35.0 [14.6, 74.4]	39.0 [20.4, 54.2]	33.9 [20.0, 45.5]	39.0 [17.3, 41.1]	23.8 [21.9, 40.1]	22.7 [11.7, 29.0]	21.7 [11.4, 26.8]	33.0 [13.1, 111.7]	18.7 [11.7, 88.8]
SStair	32.9 [20.2, 148.0]	99.4 [34.3, 146.2]	72.1 [30.5, 119.6]	39.4 [26.0, 105.4]	35.2 [11.8, 84.7]	28.6 [11.8, 54.9]	17.9 [11.4, 28.9]	12.8 [10.2, 21.4]	19.3 [11.3, 29.5]	26.3 [12.0, 58.1]
*p*-value	0.429	0.008 †	0.040 †	0.074	0.074	1.000	1.000	0.359	0.429	1.000
	Comparison of %MCRL (%, median [interquartile range]) of the BF between simulated level versus stair gait conditions									
	phase 1	phase 2	phase 3	phase 4	phase 5	phase 6	phase 7	phase 8	phase 9	phase 10
SLevel	24.6 [15.9, 39.6]	56.6 [44.4, 65.2]	71.0 [53.7, 85.4]	63.2 [60.7, 92.7]	63.9 [48.2, 73.6]	38.1 [25.4, 47.9]	29.4 [15.4, 36.9]	33.2 [19.4, 35.4]	22.1 [17.1, 27.7]	25.5 [20.6, 32.5]
SStair	27.8 [24.7, 37.9]	56.2 [48.4, 98.1]	92.6 [64.4, 121.3]	96.2 [68.1, 113.2]	66.2 [55.3, 83.7]	33.4 [24.5, 55.6]	18.9 [15.6, 40.1]	19.1 [13.0, 38.5]	25.9 [19.6, 44.1]	23.7 [20.1, 27.8]
*p*-value	0.820	0.164	0.098	0.040 †	0.910	0.429	0.820	0.910	0.359	0.429
	Comparison of %MCRL (%, median [interquartile range]) of the TA between simulated level versus stair gait conditions									
	phase 1	phase 2	phase 3	phase 4	phase 5	phase 6	phase 7	phase 8	phase 9	phase 10
SLevel	22.2 [18.2, 34.6]	31.9 [18.5, 64.2]	31.1 [22.7, 57.1]	24.6 [10.9, 47.7]	26.6 [14.9, 61.2]	35.2 [13.7, 53.8]	33.1 [23.0, 44.8]	28.9 [22.8, 58.3]	24.3 [17.4, 79.3]	27.5 [15.8, 60.1]
SStair	22.5 [19.5, 25.5]	38.6 [26.2, 64.5]	66.0 [39.0, 105.9]	52.8 [36.8, 79.5]	43.6 [11.5, 63.2]	38.2 [10.7, 60.2]	45.2 [30.0, 73.2]	30.6 [17.1, 77.9]	38.6 [11.7, 65.9]	18.2 [12.0, 29.50]
*p*-value	0.910	0.820	0.429	0.098	0.652	0.496	0.820	0.734	1.000	0.203
	Comparison of %MCRL (%, median [interquartile range]) of the medial GAST between simulated level versus stair gait conditions									
	phase 1	phase 2	phase 3	phase 4	phase 5	phase 6	phase 7	phase 8	phase 9	phase 10
SLevel	15.3 [11.4, 23.8]	17.5 [9.3, 25.5]	30.8 [10.8, 72.2]	51.7 [10.7, 72.3]	39.6 [16.0, 67.4]	33.1 [17.1, 62.1]	18.1 [12.7, 36.1]	16.4 [13.1, 37.3]	13.2 [7.0, 24.2]	14.4 [12.3, 19.3]
SStair	12.4 [8.2, 30.9]	21.0 [10.5, 31.2]	27.0 [11.2, 32.6]	24.5 [8.1, 65.9]	37.3 [13.5, 64.4]	50.9 [17.9, 57.4]	31.6 [26.1, 41.8]	20.9 [10.8, 34.8]	12.0 [8.1, 34.6]	10.7 [7.1, 30.0]
*p*-value	0.910	1.000	0.652	0.570	0.496	0.570	0.496	0.820	1.000	0.734

† indicates *p*-value < 0.05, %MCRL: (% of maximal contraction during real level walking) (10 phases per gait cycle), VL: vastus lateralis; BF: biceps femoris; TA: tibialis anterior; medial GAST: medial gastrocnemius, RLevel: real level walking condition, SLevel: simulated level walking on the robot, RStair: real stair walking condition, SStair: simulated stair walking on the robot, Wilcoxon signed-rank test on %MCRL of each of the four examined muscles during SLevel and SStair on the robot. Data are presented as median [interquartile range].

## Data Availability

The data presented in this study are available upon request from the corresponding author.

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
