# Peer review of "Comparing the Lower-Limb Muscle Activation Patterns of Simulated Walking Using an End-Effector-Type Robot with Real Level and Stair Walking in Children with Spastic Bilateral Cerebral Palsy"

_sensors, 2023, doi:10.3390/s23146579_

Round 1
Reviewer 1 Report
Dear,
please find the comments attached.
Kind regards,
the reviewer

Reviewer 2 Report
The authors study the effects of the Morning Walk robot in the rehabilitation of children. EMG signals along with the recognized gait are used to evaluate the muscle activation patterns. They compare the signals sampled from the simulated gait condition using the robot with the real gait conditions and summarize their findings. The research is appropriately planned and the data are also collected and reported in a scientific way. Mostly, I think this work is suitable to be published as an academic paper. But I am not sure if this is suitable to appear in this journal Sensors. Maybe the audiences with medical background will care more about this work.
Some suggestions:
- The authors only tested the Morning Walk robot in this work. It does not represent all robot-assisted rehabilitation and they should be cautious to conclude their findings.
- how many channels of the EMG signals are used? how do you merge them into one single vector? this is not clear to me
- in line 146-148, when the authors first introduce he RMS values, a reference should be provided
- the captions on the axis of Fig.3 is not clear
- some explanation from line 188 should be added as the caption in Table.2, similar to the head to Table.3
- too tiny fonts in fig.4~6
The paper is mostly well written. Minor editing is expected to help readers from engineering background to understand it.
Reviewer 3 Report
The manuscript titled "Comparing lower extremity muscle activation patterns of simulated walking using an end-effector-type robot with real-level and stair walking in children with spastic bilateral cerebral palsy" presents the findings of a well-designed and carefully conducted study. I have only minor comments to provide:
1. The authors should not only mention the sample size but also provide information about the sampling frame.
2. It is necessary to clarify who diagnosed cerebral palsy and the diagnostic criteria used.
3. In Tables 1-4, not all abbreviations are explained. Please provide explanations for all abbreviations used.
4. It would be beneficial to include a subsection on Statistical Analysis that describes the methods used for statistical analyses.
Reviewer 4 Report
The article entitled “Comparing lower extremity muscle activation patterns of simulated walking using an end-effector-type robot with real-level and stair walking in children with spastic bilateral cerebral palsy” This study aimed to compare muscle activation patterns of real-level and stair walking with those of walking simulated with an end-effector-type robot in children with spastic cerebral palsy
Below are some suggestions:
In the Abstract:
- The abstract is well written and objective, however a suggestion is to insert an introductory opening paragraph citing cerebral palsy and its importance.
1. In the Introduction:
Regarding the introduction:
- I suggest the insertion of an abstract graphic demonstrating the design of the study in a clear way that becomes attractive to the reader;
- the authors can insert a writing about paralysis, its importance and consequences in the children's lives, emphasizing the clinical potential of the article.
2. Materials and Methods:
- The methodology should be more organized, identifying items, for example: 2.1 Participants* In general, I believe that the methodology needs to be better described;
- Table 1 requires data adjustment as it is difficult to visualize;
- Better describe the data devices;
- Figure 5 needs to be improved, it is very poor quality (an experimental design can be done).
3. Results
- All tables must be adjusted, it is difficult to visualize the data;
- I also suggest dividing the graphics, impossible to observe the data.
4. Discussion
- Authors should start the discussion by inserting a first paragraph summarizing what was done, as well as the main result;
- The discussion is disorganized.
5. Conclusion
- The conclusion is well written, bringing a summary of what was done in the research and future perspectives, demonstrating the clinical applicability.
* Authors must adjust the references according to the journal's guidelines, as well as insert a greater number of citations. The number of references is insufficient.
I suggest moderate revisions in the English language.
Round 2
Reviewer 4 Report
Thanks to the authors who made all the suggestions in the manuscript.
Minor editing of English language required